# The Association of Vitamin D Receptor Gene Polymorphisms with Vitamin D, Total IgE, and Blood Eosinophils in Patients with Atopy

**DOI:** 10.3390/biom14020212

**Published:** 2024-02-11

**Authors:** Daina Bastyte, Laura Tamasauskiene, Ieva Stakaitiene, Rasa Ugenskiene, Brigita Gradauskiene (Sitkauskiene)

**Affiliations:** 1Department of Immunology and Allergology, Lithuanian University of Health Sciences, LT-50161 Kaunas, Lithuania; 2Laboratory of Immunology, Department of Immunology and Allergology, Lithuanian University of Health Sciences, LT-50161 Kaunas, Lithuania; 3Department of Genetics and Molecular Medicine, Lithuanian University of Health Sciences, LT-50161 Kaunas, Lithuania

**Keywords:** atopy, asthma, atopic dermatitis, vitamin D, vitamin D deficiency, vitamin D receptor, single nucleotide polymorphism

## Abstract

Background: In order to improve the control of atopic diseases, it is important to clarify the pathogenesis of atopy and identify its various triggers. Single nucleotide polymorphisms (SNPs) of the vitamin D receptor gene (*VDR*) may impact atopy. The aim of this study was to investigate the possible associations between *VDR* SNPs and vitamin D, total IgE, and eosinophils in atopy. Methods: In total, 203 adults, including 122 patients with atopic diseases (45 with atopic dermatitis, 77 with allergic asthma) and 81 healthy controls, were involved in the study. The blood eosinophil count was determined with an automated hematology analyzer. Vitamin D and total immunoglobulin E (IgE) levels were evaluated using the ELISA method. Polymorphisms in the *VDR* gene were analyzed with real-time PCR using TaqMan probes. Results: We analyzed six *VDR* single nucleotide polymorphisms and found a significant association between *VDR* rs731236 GG genotype and normal vitamin D levels in atopic patients and healthy subjects (OR 11.33; 95% CI: 1.049–122.388 and OR 4.04; 95% CI: 1.117–14.588, respectively, *p* < 0.05). Additionally, the study results revealed a significant relationship between the *VDR* rs2228570 GG genotype and normal vitamin D levels in patients with atopy and healthy subjects (OR 3.80; 95% CI: 1.190–12.134 and OR 2.09; 95% CI: 1.044–4.194, respectively, *p* < 0.05). The rs2228570 allele A was associated with decreased vitamin D levels in patients with atopy and healthy subjects (OR 0.28; 95% CI: 0.098–0.804 and OR 0.229; 95% CI: 0.069–0.761, respectively, *p* < 0.05). The *VDR* rs3847987 genotypes AA and AC were significantly associated with normal vitamin D levels in healthy subjects (OR 35.99; 95% CI: 6.401–202.446 and OR 4.72; 95% CI: 1.489–15.007, respectively, *p* < 0.05). In addition, a decreased amount of vitamin D was associated with atopic diseases such as atopic dermatitis and allergic asthma (OR 0.49; 95% CI: 0.439–1.308 and OR 0.58; 95% CI: 0.372–0.908, respectively, *p* < 0.05). The rs11168293 allele T was associated with the normal range of total IgE in atopy (OR 2.366; 95% CI: 1.133–5.027; *p* < 0.05). Significant associations were found between *VDR* rs731263 allele G, rs11168293 allele G, and increased blood eosinophil levels in patients with atopy (OR 0.319; 95% CI: 0.163–0.934 and OR 0.323; 95% CI: 0.112–0.935, respectively, *p* < 0.05). Conclusions: A decreased vitamin D level showed a significant relationship with atopic diseases (atopic dermatitis and allergic asthma). The association between the *VDR* gene polymorphisms rs2228570, rs731236, and rs11168293 and vitamin D, total IgE, and blood eosinophils in patients with atopy suggested that *VDR* polymorphisms and the vitamin D level should be considered when examining the factors associated with atopy.

## 1. Introduction

Atopic diseases, including atopic dermatitis (AD), allergic asthma (AA), and allergic rhinitis (AR), have seen a significant increase in prevalence over the past few decades. These conditions are characterized by an exaggerated immune response to common environmental allergens, leading to chronic inflammation and a variety of symptoms. These conditions now affect approximately 20% of the global population [1]. The increasing prevalence of atopic diseases highlights the need for further research to understand their complex etiology and to develop more effective prevention and treatment strategies. However, while atopic diseases impact both children and adults worldwide, their pathogenesis has still not been clearly established.

Major atopic diseases share the same atopic background, characterized by aberrant cell-mediated immunity and overproduction of immunoglobulin E (IgE) [2]. This heightened reactivity results in inflammation, a hallmark of atopic conditions, and manifests in a spectrum of symptoms. Individuals with atopic diseases may experience itching, redness, respiratory distress, and other manifestations, depending on the specific condition. The complex development of these diseases is associated with a multitude of factors [3,4]. Genetic predisposition, environmental exposures to allergens, and immune system dysregulation are among the various factors that have been linked to the onset and progression of these atopic conditions. Recognizing the interconnected nature of these elements is crucial for advancing our understanding and developing comprehensive approaches to the prevention and treatment of major atopic diseases.

Additionally, vitamin D deficiency is common worldwide [5], and there is a possibility that vitamin D may be partly related to the increased prevalence of atopic diseases in recent years [6]. Vitamin D is believed to contribute to immune system regulation and may play a role in modulating the inflammatory responses associated with allergic reactions. It is important to note that most of the circulating vitamin D forms act by binding to a vitamin D receptor (*VDR*), and the single nucleotide polymorphisms (SNPs) of *VDRs* may affect the occurrence of atopy by regulating *VDR* expression and the vitamin D level [3]. Moreover, certain *VDR* polymorphisms may be associated with alterations in vitamin D levels and immune responses. VDR genetic variations may affect the efficiency of vitamin D binding to receptors, potentially influencing downstream signaling pathways involved in immune system modulation [3]. As a result, individuals with specific *VDR* polymorphisms might exhibit variations in their susceptibility to various diseases, including atopic diseases like atopic dermatitis, allergic asthma, and allergic rhinitis. Studies have shown that *VDR* gene polymorphisms, such as rs7975232, rs1544410, and rs731236, may confer susceptibility to allergic diseases in certain populations [7]. Furthermore, several studies have demonstrated a significant association between *VDR* gene polymorphisms and asthma risk [8], or have indicated that rs2228570 could be associated with the risk of atopic dermatitis [9]. On the other hand, there is controversial data from various studies regarding the association of *VDR* SNPs with atopic conditions [10]. Moreover, certain *VDR* SNPs, for example, rs11168293, remain poorly explored. Understanding the intricate relationship between genetic variations within the *VDR* gene, vitamin D metabolism, and immune responses is crucial for unraveling the underlying mechanisms contributing to disease susceptibility and developing targeted approaches for disease prevention and management.

In order to gain deeper insights into the potential impact of *VDR* SNPs on the development of atopic diseases, it is essential to examine their association with markers of atopy. Existing studies have proposed that *VDRs* may exert an influence on the immune system by regulating the production or activity of various immune cells, including eosinophils, as well as modulating IgE production by B cells. It was determined that vitamin D, through the *VDR*, helps maintain low serum IgE responses [11]. Several clinical studies have shown that vitamin D may be associated with IgE levels and eosinophilic inflammation in subjects with atopy [6,12]. However, the relationships between *VDR* SNPs, IgE, and eosinophilia in atopy have not been explored in-depth. How the vitamin D and *VDR* signaling pathway affects the production of IgE antibodies, leads to the activation of eosinophils, and triggers the overall allergic response has still not been clearly established.

There is a possibility that specific *VDR* SNPs may be associated with a discernible impact on vitamin D levels. Individuals with these specific *VDR* SNPs may be more prone to experiencing higher rates of hypovitaminosis D. Moreover, we hypothesize that vitamin D deficiency, linked to the *VDR* SNPs, correlates with increased severity of atopy. This hypothesis suggests a genetic basis for the interplay between specific SNPs, vitamin D status, and the manifestation of heightened atopic responses, as evidenced by total IgE and blood eosinophils in individuals with atopic conditions compared to healthy subjects. Evaluating the associations between 25-hydroxyvitamin D (25(OH)D) levels and *VDR* SNPs in patients with atopic diseases and healthy subjects could help us to understand the role of *VDR* in atopy and elucidate its effects on the variation of vitamin D levels.

Therefore, the present study aimed to investigate SNPs in *VDR* genes and the vitamin D level in patients with atopic diseases and healthy individuals, and to analyze their associations with different parameters of atopy (total IgE, blood eosinophils).

## 2. Materials and Methods

### 2.1. Study Design

This case–control study was carried out at the Hospital of the Lithuanian University of Health Sciences Kauno Klinikos. The study was approved by the ethics committee (No. BE-2-74), and informed consent was obtained from all the participants. A total of 203 adults were accepted to participate and had their peripheral blood samples collected. There were 122 patients with mild to moderate persistent atopic diseases (45 with atopic dermatitis, 77 with allergic asthma) and 81 healthy subjects as controls. To ensure that the results of the study were not influenced by vitamin D supplements, only subjects who had not taken vitamin D supplements for at least three months were included in the study. Blood samples were collected during the period of September 2020 to April 2023. All patients represented the same ethnic group of native Lithuanians. Patients with atopic dermatitis were diagnosed according to the criteria of Hanifin and Rajka [13]. Asthmatic subjects were diagnosed and classified according to the Global Initiative for Asthma (GINA) recommendations [14]. Patients with atopic dermatitis underwent assessment using the SCORAD index. Asthmatic patients underwent tests for forced vital capacity (FVC) and forced expiratory volume in 1 s (FEV1). No hospitalization due to atopic diseases during the study was recorded. The inclusion criteria were patients with mild to moderate persistent asthma or atopic dermatitis with allergic sensitization, aged 18 years or older. Exclusion criteria were as follows: any acute or chronic respiratory diseases (except asthma), pregnancy, and autoimmune and oncologic diseases. The control group was composed of randomly selected healthy patients who were not sensitized to allergens.

### 2.2. Sample Collection and Storage

Peripheral venous blood samples were collected from the subjects in K3 EDTA tubes for investigation of *VDR* polymorphisms and for the eosinophil count. Samples for the total IgE and vitamin D assay were drawn into serum tubes. The blood specimens were given a personal identifier number that was used to link and maintain the biological information derived from them. Serum tubes were centrifuged at 3500 rpm for 10 min, and the serum was separated and frozen at −80 °C for further analysis.

### 2.3. DNA Extraction and SNP Genotyping

DNA was isolated using the QIAamp DNA blood mini kit (Qiagen, Hilden, Germany) according to the manufacturer’s instructions. Six polymorphisms in the *VDR* (rs7975232, rs1544410, rs731236, rs3847987, rs2228570, rs11168293) gene in chromosome region 12q13.11 were analyzed using TaqMan SNP Genotyping Assays (Thermo Fisher Scientific, Waltham, CA, USA), according to the manufacturer’s protocol.

### 2.4. Evaluation of Serum Vitamin D, Total IgE, and Blood Eosinophils

Measurements of serum 25(OH)D, representing vitamin D levels, were performed by using the enzyme-linked immunosorbent assay ELISA using a commercial kit (BioVendor, Brno, Czech Republic). The inter-assay CV for 25(OH)D was 20%. The limit of detection was 2.81 ng/mL. Data analysis was performed according to vitamin D content: deficiency < 20 ng/mL (<50 nmol/L), insufficiency 20–30 ng/mL (50–75 nmol/L), normal 30–50 ng/mL (75–125 nmol/L) [15,16,17,18].

Blood eosinophil evaluation was performed with an automated hematology analyzer (Sysmex, Kobe, Japan). Elevation of blood eosinophils was defined as ≥5% of total leukocytes [19].

Measurements of total IgE levels in serum were performed using an ELISA commercial kit (IBL International, Hamburg, Germany). The average inter-assay CV was 4.1%. The limit of detection was 0.8 IU/mL. According to the manufacturer’s recommendations, the normal range for total IgE in adults was less than 100 IU/mL [20].

### 2.5. Statistical Analysis

Statistical analysis was performed using the SPSS Statistics software, version 29 (IBM Corp., Armonk, NY, USA), and Microsoft Excel (Microsoft, Redmond, WA, USA). Methods of statistical analysis were selected after performing the Kolmogorov–Smirnov test. The significance of the differences between the studied groups was analyzed using the Student *t*-test (means, Gaussian populations), Mann–Whitney U test (medians, non-Gaussian populations), ANOVA test (means, Gaussian populations), or Kruskal–Wallis test (medians, non-Gaussian populations). The distributions of *VDR* gene polymorphisms between groups were assessed using the chi-squared test. Binary logistic regression was used to evaluate the impact of the studied genotypes for vitamin D and markers of atopy. The odds ratios (ODs) were calculated from frequency with a 95% confidence interval (95% CI). Results were statistically significant when p was less than 0.05.

## 3. Results

### 3.1. Characteristics of the Studied Subject

The general characteristics of the study groups are summarized in Table 1. The studied groups did not differ significantly according to their sex and age. The total IgE levels and blood eosinophil count were significantly higher in subjects with atopic diseases than in the control group (Table 1). There were no significant differences in total IgE and blood eosinophils between subjects with different atopic diseases (patients with atopic dermatitis and allergic asthma).

### 3.2. Distribution of VDR Gene Polymorphisms in the Studied Groups

To analyze the associations of *VDR* polymorphisms with atopy, we divided the study subjects into atopic and control groups. The distribution of *VDR* SNPs between atopic patients and the control group is demonstrated in Table 2. No statistically significant differences in genotype distribution were found between patients with atopy and the control group.

### 3.3. Relationship between VDR Polymorphisms and Vitamin D Level

The comparison of vitamin D blood levels in the studied groups is demonstrated in Table 3. Binary logistic regression analysis showed that decreased vitamin D levels were associated with atopic diseases such as atopic dermatitis or allergic asthma (Table 3). In Table 4 and Table 5, *VDR* polymorphisms were compared in larger groups of subjects with atopic diseases divided based on their vitamin D levels: one group with normal vitamin D levels (>30 ng/mL) and another with vitamin D deficiency (<30 ng/mL). The results showed a significant relationship between several *VDR* gene SNPs and vitamin D levels in patients with atopic diseases (Table 4 and Table 5). A tendency was observed that the VDR rs731236 genotype AA was more common in studied groups with decreased vitamin D levels, compared to the rs731236 AA genotype in subjects with normal vitamin D levels (Table 4). Moreover, results showed a significant association between the VDR rs731236 GG genotype and normal vitamin D levels in atopic patients and healthy subjects (Table 5). The VDR rs2228570 genotype GG was more frequent in atopic patients with normal vitamin D levels, compared to atopic patients with decreased vitamin D levels (Table 4). Additionally, the study results revealed a significant relationship between the *VDR* rs2228570 GG genotype and normal vitamin D levels in patients with atopy and healthy subjects (Table 5). The rs2228570 allele A was associated with decreased vitamin D levels in patients with atopy and healthy subjects (Table 5). Moreover, the VDR rs3847987 CC genotype was statistically more frequent in healthy subjects with lower vitamin D levels compared to those with normal vitamin D levels, while the rs3847987 AC genotype was more common in healthy subjects with normal vitamin D levels compared to controls with decreased vitamin D levels (Table 4). It is important that the rs3847987 genotypes AA and AC and allele A were significantly associated with normal vitamin D levels in healthy subjects, but not in patients with atopic diseases (Table 5).

### 3.4. Relationship between VDR Polymorphisms and Markers of Atopy (Total IgE, Blood Eosinophils)

Atopic patients with atopic dermatitis and allergic asthma exhibited significantly higher total IgE and blood eosinophil levels compared to the control group. No significant differences were observed between the different atopic groups (Table 1). To analyze the associations of *VDR* polymorphisms with total IgE and blood eosinophil levels in atopic diseases, we divided atopic patients into groups based on total IgE levels (>100 IU/mL and ≤100 IU/mL) and their eosinophil percentage from total leukocytes (>5% and ≤5%). The genotype distribution between the studied groups is given in Table 6. The results of logistic regression analysis on the association of *VDR* SNPs with IgE and eosinophils in atopic patients are shown in Table 7 and Table 8. We found a significant relationship between several *VDR* SNPs and total IgE and blood eosinophil levels in atopic patients. The logistic regression analysis in the *VDR* polymorphisms allelic model showed that the minor allele T of *VDR* rs11168293 was associated with a normal range of total IgE levels (<100 IU/mL) in patients with atopy (Table 7). Atopic patients carrying the *VDR* rs11168293 GG genotype had significantly higher IgE levels, compared to those with the GT genotype (Table 7). Among the atopic patients carrying the *VDR* rs11168293 genotype TT, 26.0% had blood eosinophil levels lower than 5% of all leukocytes, in contrast to 10.2% of patients with the rs11168293 genotype TT and increased eosinophil levels (Table 6). Moreover, a significant correlation was found between the *VDR* rs11168293 GG genotype and allele G and increased blood eosinophil levels in atopic patients (Table 8). Additionally, 37.1% of atopic patients with the VDR rs731236 AA genotype exhibited normal blood eosinophil levels, compared to 18.8% of patients with the AA genotype and increased blood eosinophil levels (Table 6). The results showed a significant relationship between the *VDR* rs731236 AA genotype and normal blood eosinophil levels, compared to the AG genotype, in atopic patients. Allele G of *VDR* rs731236 was linked to increased blood eosinophil levels in patients with atopy (Table 8).

### 3.5. Multifactorial Analysis

Multifactorial analysis was conducted to examine the impact of decreased vitamin D levels in combination with the analyzed *VDR* gene polymorphisms on increased IgE and blood eosinophil levels in atopic diseases (Table 9). A significant association was found between high total IgE and blood eosinophil levels in atopy for patients with a combination of low vitamin D levels and the *VDR* rs11168293 GG genotype. Moreover, vitamin D levels, the rs11168293 GG genotype, and the rs731236 G allele combination were significantly associated with blood eosinophils but not with total IgE. The data showed no significant association between insufficient vitamin D levels and high IgE or eosinophil levels in patients with atopy.

## 4. Discussion

This case–control study investigated the relationship between *VDR* SNPs, vitamin D, and markers of atopy in Lithuanian adults with atopic diseases (atopic dermatitis and allergic asthma). Six *VDR* single nucleotide polymorphisms were explored, and the analysis showed a significant link between *VDR* rs731236 and rs2228570 and vitamin D levels in subjects with atopy and healthy controls. The rs3847987 SNP was significantly associated with vitamin D levels in healthy subjects but not in atopic patients. *VDR* SNP rs11168293 was associated with total IgE levels and two others, *VDR* SNP’s rs11168293 and rs731236, were related to eosinophils in atopy.

The results of the present study indicated a significant association between decreased vitamin D levels and atopic diseases such as atopic dermatitis and allergic asthma. Furthermore, several studies have shown that the vitamin D status during atopy could be associated with the risk or severity of atopic diseases [21,22,23,24]. Assuming that vitamin D has an anti-inflammatory effect and plays an important role in immune system regulation [25], evidence suggests that it has an involvement in balancing the ratio of T helper 1 cells (Th1) to Th2, as well as in modulating the immune balance between Th17 cells and T regulatory cells [26,27]. Atopic diseases might be linked to low vitamin D levels, especially in individuals with mutations in *VDR* genes [6]. There is a possibility that genetic variations within the *VDR* gene play a significant role in human health, impacting vitamin D metabolism and susceptibility to various diseases. Certain *VDR* polymorphisms might influence vitamin D levels, immune responses, and the risk of developing atopic diseases [28]. The present study revealed that the rs2228570 genotype GG was associated with normal vitamin D levels, whereas the rs2228570 allele A was associated with decreased vitamin D levels in both patients with atopy and healthy subjects. There is a possibility that certain *VDR* SNPs may affect the structure of gene-expressing proteins and influence the function of the *VDR* signaling pathway in vitamin D metabolism [29,30]. For example, the rs2228570 polymorphism, located at the 5′ end of exon 2 of the *VDR* gene, might affect mRNA splicing or impact the *VDR* protein structure [30]. Studies have suggested that changing from the C allele to the T allele in rs2228570 might result in the production of a slightly shorter *VDR* protein [31]. The shorter *VDR* form could be less efficient in regulating gene expression in response to vitamin D [32]. Tuncel et al. investigated the four most common *VDR* SNPs, rs1544410, rs731236, rs7975232, and rs2228570, in a cohort of Turkish Cypriots and found significant associations between the rs2228570 T allele and decreased vitamin D levels [33]. Several publications have reported associations between *VDR* gene polymorphisms, such as rs1544410 or rs2228570, and decreased vitamin D levels [33,34,35]. Additionally, rs2228570 has been associated with an increased risk of atopic dermatitis or allergic asthma in some studies [9,36,37]. A recent report even showed that the rs2228570 A allele increased the risk of allergic rhinitis [38].The results of our study support that the rs2228570 allele A is associated with decreased vitamin D levels in both patients with atopy and healthy subjects. Moreover, in this study, a significant association was found between the VDR rs731236 genotype GG and normal vitamin D levels in the studied patients with atopy and the control group. On top of this, Irish scientists revealed an association between the C allele of rs731236 and the risk of uncontrolled asthma [39]. Moreover, the present study revealed a significant association between the rs3847987 genotypes AA and AC and allele A and normal vitamin D levels in healthy controls but not patients with atopic diseases. These results suggest that *VDR* SNP rs3847987 may not be related to vitamin D in atopic diseases. However, case–control and family-based studies in China demonstrated that *VDR* rs3847987 may be associated with obesity [40] and hypertension [41]. Moreover, rs3847987 was reported to be associated with vitamin D levels [41,42]. Unfortunately, our study did not evaluate the association of vitamin D and *VDR* SNPs with indicators such as body mass index (BMI), hypertension, vitamin D supplementation, or seasonality; therefore, further studies are needed. However, our data suggested that the *VDR* polymorphisms rs731236, rs2228570, and rs3847987 might be significantly associated with vitamin D levels in healthy subjects. Moreover, the *VDR* SNPs rs731236 and rs2228570 could influence the relationship between low vitamin D levels and atopic diseases. It is important to note that the conclusions of published studies have not been consistent, and the association between these *VDR* polymorphisms and vitamin D levels has not been entirely established across different populations.

This study found a significant relationship between the *VDR* rs11168293 allele T and the normal range of total IgE levels (<100 IU/mL). Additionally, the rs11168293 genotype GG was associated with increased IgE levels in atopy. Moreover, the results revealed a significant link between the rs11168293 allele G and increased levels of eosinophils in atopic patients. Other studies have reported a statistically significant association between the *VDR* SNPs rs2239185, rs 7975232, and rs731236 and evaluated IgE levels [28,36]. However, we found no significant association between other *VDR* SNPs. IgE is produced by B lymphocytes [11]. Therefore, we hypothesized that rs11168293 may affect B cell functions through the T allele and eosinophilic inflammation through the G allele. This also supports the hypothesis that vitamin D and *VDR* SNPs may disturb Th2 cell responses and promote the development of atopy [30]. On the other hand, we found no statistically significant results between rs11168293 SNPs and vitamin D levels in the studied groups. There is a possibility that rs11168293 may be independently associated with atopic outcomes, particularly in relation to total IgE levels and blood eosinophils. This suggests that specific *VDR* SNPs may directly influence the immune response associated with atopic conditions. Therefore, this hypothesis implies that the impact of *VDR* polymorphisms on atopic outcomes may be significant, even when not accounting for factors such as vitamin D levels. Unfortunately, the rs11168293 variant is poorly studied, and to our knowledge, there are no published articles on the role of *VDR* rs11168293 in the context of atopic diseases. The *VDR* gene rs11168293 variant needs to be further explored to confirm our results.

Of the six identified SNPs, the rs11168293 and rs731236 SNPs were associated with blood eosinophil levels in atopic subjects. Logistic regression analysis revealed that the G allele of rs731236 and G allele of rs11168293 were linked with increased blood eosinophil levels in atopic diseases. Furthermore, the rs731236 AA genotype and rs11168293 TT genotype were associated with blood eosinophil levels < 5%. Eosinophils, as part of the immune system, play a role in the allergic response and are often elevated in atopy [43]. *VDRs* prolong the survival of eosinophils and enhance the expression of membrane receptors, inhibiting their apoptosis [44]. Moreover, our findings support the hypothesis that vitamin D, acting through the *VDR*, can influence the immunological cascade by suppressing the response of T2-high lymphocytes and reducing the production of IL-5 [45]. Decreased IL-5 levels translate to a noteworthy impact on eosinophil counts, as IL-5 is a key regulator of eosinophil growth, maturation, and activation [40]. In addition, experimental studies in mice have demonstrated that *VDR* deficiency is a causative factor for spontaneous activation of eosinophils [46]. Chinese scientists have revealed that *VDRs* contribute to maintaining eosinophil homeostasis by regulating the gene transcription of eosinophil mediators [46]. Furthermore, a publication on *VDR* SNPs in obese Egyptian men reported findings indicating a statistically significant association between the VDR polymorphism rs1544410 and eosinophil counts, although no significant association was observed with vitamin D levels [47]. In contrast, we found no significant association between rs1544410 and eosinophils; however, indicators such as body mass index were not analyzed in our study. However, our findings regarding the association between *VDR* SNPs rs11168293 and rs731236 and eosinophil count in atopic patients lead us to propose the hypothesis that *VDRs* may influence eosinophil count and immunological inflammation. On top of this, several studies have shown an association between vitamin D deficiency and increased blood eosinophils in patients with asthma or allergic rhinitis [12,48,49]. As mentioned before, this study showed that rs731236 and rs2228570 were related to vitamin D levels in subjects with atopic diseases. Furthermore, the combination of low vitamin D levels, the rs11168293 GG genotype, and the rs731236 G allele was significantly associated with blood eosinophil levels. The data of this study suggested that the vitamin D–*VDR*-eosinophil axis may play a very important role in the pathogenesis of atopic diseases. There is a possibility that variations in vitamin D levels and the functionality of the specific *VDR* SNPs may contribute to the dysregulation of eosinophil activity, thereby influencing the development and exacerbation of atopic conditions. This hypothesis suggests a dynamic interplay between vitamin D, its receptor, and eosinophils, where alterations in this axis may contribute to the immunological processes underlying atopic diseases.

The strength of this study lies in its detailed exploration of vitamin D polymorphisms in patients with atopic diseases compared to the control group. It represents a valuable contribution to the understanding of genetic influences on atopic diseases. The regulatory pathways in the pathogenesis of atopic diseases, involving vitamin D, *VDR* genetic variations, eosinophils, and IgE, underscore the multifaceted nature of vitamin D’s role in immune modulation. This understanding presents potential avenues for therapeutic interventions to mitigate conditions associated with elevated eosinophil counts and IgE levels in atopic diseases. The exploration of *VDR* polymorphisms in the context of atopy opens up avenues for personalized medicine and could be the basis for new diagnostic methods of atopic diseases. Understanding how genetic variations influence responses to vitamin D and, consequently, immune function provides a foundation for tailoring interventions based on an individual’s genetic profile. Moreover, our study represents pioneering work in Lithuanian atopic patients, and we hope that it will open new horizons for future research in this area.

Our study has some limitations. First, we explored statistical associations, and thus further research regarding the underlying mechanisms is required. Genetic variants, those related to the *VDR* gene, were a primary focus of our study. In addition, genetic variants alone may not be enough to induce atopy and atopic diseases; the interaction between genes and environmental factors should not be ignored. While we identified associations between specific *VDR* SNPs, vitamin D, and atopic markers, the causal relationships and the direction of causality remain ambiguous. The identified associations should be considered as initial findings, requiring more in-depth investigation. Blood samples from studied patients were collected during the period from September to April. However, we may not have included all potential confounding factors in the analyses, such as environmental factors like sun exposure, seasonal vitamin D variation, and diet, as well as nutritional status parameters like vitamin D nutritional intake, body weight and BMI, or interactions of *VDR* genes with other genes involved in vitamin D metabolism in atopy that may play an additional role. Future studies should explore these interactions to obtain a more comprehensive understanding. Notably, specific atopic outcomes, including measures such as inhaler use and hospitalization, were not included in our study. The exclusion of these crucial indicators may limit the comprehensiveness of our findings. Moreover, atopic diseases exhibit dynamic and multifaceted characteristics. Our study provides a snapshot of certain aspects, but the dynamic nature of atopic diseases requires continuous observation and exploration. Future research should consider longitudinal approaches and capture the evolving nature of atopic diseases over time. This study covers only the population of Lithuania. The unique genetic, environmental, and cultural characteristics of this population may influence the generalizability of our findings. To enhance the external validity of our results, it is essential to replicate the study in diverse populations with varying genetic and environmental contexts. In addition, there is limited available research data in European populations, so the present findings should be verified in other population settings. Future studies are required to analyze the role of *VDR* gene polymorphisms in the immune response of atopic diseases in different populations.

## 5. Conclusions

The study results showed a relationship between decreased vitamin D levels and atopic diseases. Additionally, there was a significant association between *VDR* single nucleotide polymorphisms, specifically the rs731236 GG and rs2228570 GG genotypes and normal vitamin D levels, the rs2228570 allele A and a lower vitamin D level than normal in atopic patients and healthy subjects, the rs11168293 GG genotype and rs731236 allele G and increased amounts of blood eosinophils, and the rs11168293 genotype GG and an increase in the total IgE level in atopic diseases. These findings suggested that the *VDR* single nucleotide polymorphisms rs2228570, rs731236, and rs11168293 and vitamin D status should be considered when examining the factors associated with atopy.

## Figures and Tables

**Table 1 biomolecules-14-00212-t001:** General characteristics of the study groups.

	Subjects with Atopy (*n* = 122)		Control Group (*n* = 81)
	Atopic Dermatitis (*n* = 45)	Allergic Asthma (*n* = 77)	
Male/female, N	18/27	27/50	32/49
Age, years	31.52 ± 1.52	39.52 ± 1.57	35.57 ± 1.45
Total IgE IU/mL	767.19 ± 232.16 *	689.72 ± 169.59 *	24.99 ± 5.68
Blood eosinophil count (10^9^/L)	0.39 ± 0.07 **	0.35 ± 0.07 **	0.06 ± 0.01
Blood eosinophils (% from leukocytes)	5.55 ± 0.76 **	5.59 ± 0.54 **	1.09 ± 0.15
SCORAD	38.29 ± 4.11	N/A	N/A
Lung function:			
FEV 1 (% from predicted value)	N/A	94.87 ± 1.92	N/A
FEV1 (l)	N/A	3.18 ± 0.11	N/A
FVC (% from predicted value)	N/A	101.89 ± 2.03	N/A
FVC (l)	N/A	4.06 ± 0.14	N/A
FEV1/FVC (ratio)	N/A	0.84 ± 0.17	N/A

Significant difference compared to the control group: * *p*-value < 0.05; ** *p*-value < 0.001. Values are presented as mean ± SEM unless otherwise indicated. SCORAD—Severity Scoring of Atopic Dermatitis index; FEV1—forced expiratory volume in one second; FVC—forced vital capacity.

**Table 2 biomolecules-14-00212-t002:** Distribution of polymorphisms in *VDR* gene in atopic (*n* = 122) and control (*n* = 81) groups.

Gene	SNP	Group	Genotype Frequency (%)			*p*-Value	MAF	*p*-Value
*VDR*	rs731236 (TaqI) A > G		AA	AG	GG	0.835	G	NS
		Atopy	29.7	60.2	10.2		0.403	
		Control	26.0	62.3	11.7		0.428	
	rs7975232 (ApaI) A > C		AA	AC	CC	0.893	C	NS
		Atopy	25.4	50.0	24.6		0.496	
		Control	25.9	46.9	27.2		0.494	
	rs1544410 (BsmI) C > T		CC	TC	TT	0.808	T	NS
		Atopy	41.8	48.4	9.8		0.277	
		Control	39.2	48.1	12.7		0.367	
	rs2228570 (FokI) G > A		GG	AG	AA	0.650	A	NS
		Atopy	32.8	50.8	16.4		0.421	
		Control	27.2	53.1	19.8		0.462	
	rs3847987 C > A		CC	CA	AA	0.646	A	NS
		Atopy	53.3	43.4	3.3		0.250	
		Control	55.6	43.2	1.2		0.226	
	rs11168293 G > T		GG	GT	TT	0.672	T	NS
		Atopy	40.2	40.2	19.7		0.397	
		Control	42.0	43.2	14.8		0.364	

MAF—minor allele frequency. NS—not significant.

**Table 3 biomolecules-14-00212-t003:** Atopic diseases and vitamin D level.

Vitamin D Level (ng/mL)	Groups *n* (%)		OR	95% CI	*p*-Value
<20 20–30 30–50	Atopic dermatitis (*n* = 45)	Control (*n* = 81)			
	20 (44.4%)	31 (38.3%) 32 (39.5%) 18 (22.2%)	1.076	0.481–2.410	0.858
	18 (40.0%)				
	7 (15.6%)		0.758	0.439–1.308	0.319
	Allergic asthma (*n* = 77)				
	51 (66.2%)		0.304	0.144–0.642	0.002 *
	16 (20.8%)				
	10 (13.0%)		0.581	0.372–0.908	0.017 *

* Significant difference. OR: odds ratio. CI: confidence interval.

**Table 4 biomolecules-14-00212-t004:** Distribution of *VDR* genotypes in the studied groups according to vitamin D level.

SNP	Vitamin D Level (ng/mL)	Genotypes *n* (%)			*p*-Value
rs731236 (TaqI) A > G	Atopic group	GG	AG	AA	
	>30	3 (17.6)	13 (76.5)	1 (5.9)	
	≤30	9 (8.9)	58 (57.4)	34 (33.7)	0.056
	Control group				
	>30	3 (16.7)	12 (66.7)	3 (16.7)	
	≤30	6 (9.5)	36 (57.1)	17 (27.0)	NS
rs7975232 (ApaI) A > C	Atopic group	AA	AC	CC	
	>30	6 (35.3)	9 (52.9)	2 (11.8)	
	≤30	25 (23.8)	52 (49.5)	28 (26.7)	NS
	Control group				
	>30	5 (27.8)	10 (55.6)	3 (16.7)	
	≤30	16 (25.4)	28 (44.4)	19 (30.2)	NS
rs1544410 (BsmI) C > T	Atopic group	CC	CT	TT	
	>30	3 (17.6)	11 (64.7)	3 (17.6)	
	≤30	48 (45.7)	48 (45.7)	9 (8.6)	NS
	Control group				
	>30	7 (38.9)	7 (38.9)	4 (22.2)	
	≤30	24 (38.1)	31 (49.2)	6 (9.5)	NS
rs2228570 (FokI) G > A	Atopic group	GG	AG	AA	
	>30	10 (58.8)	5 (29.4)	2 (11.8)	0.047 *
	≤30	30 (28.6)	57 (54.3)	18 (17.1)	
	Control group				
	>30	3 (16.7)	8 (44.4)	7 (38.9)	NS
	≤30	19 (30.2)	35 (55.6)	9 (14.3)	
rs3847987 C > A	Atopic group	CC	AC	AA	
	>30	7 (41.2)	9 (52.9)	1 (5.9)	NS
	≤30	58 (55.2)	43 (41.0)	4 (3.8)	
	Control group				
	>30	5 (27.8)	13 (72.2)	0 (0)	0.018 *
	≤30	40 (63.5)	22 (34.9)	1 (1.6)	
rs11168293 G > T	Atopic group	GG	GT	TT	
	>30	5 (29.4)	8 (47.1)	4 (23.5)	NS
	≤30	44 (41.9)	41 (39.0)	20 (19.0)	
	Control group				
	>30	7 (38.9)	10 (55.6)	1 (5.6)	NS
	≤30	27 (42.9)	25 (39.7))	11 (17.5)	

* Significant difference. NS—not significant.

**Table 5 biomolecules-14-00212-t005:** Association between *VDR* gene polymorphisms and vitamin D level (<30 ng/mL) in patients with atopic diseases.

Gene	SNP	Genotype and Allelic Model	Group	OR	95% CI	*p*-Value
*VDR*	rs731236 (TaqI) A > G	GG vs. AA AG vs. AA	Atopy	11.333	1.049–122.388	0.046 *
				7.621	0.954–60.850	0.055
			Control	4.037	1.117–14.588	0.033 *
				2.182	0.957–4.976	0.064
		G carrier vs. G non-carrier A carrier vs. A non-carrier	Atopy	8.119	1.033–63.833	NS
				0.457	0.110–1.893	NS
			Control	2.024	0.519–7.898	NS
				1.350	0.318–5.726	NS
	rs7975232 (ApaI) A > C	CC vs. AA AC vs. AA	Atopy	1.833	0.788–4.265	NS
				1.387	0.444–4.326	NS
			Control	0.784	0.415–1.480	NS
				2.399	0.820–7.019	NS
		A carrier vs. A non-carrier C carrier vs. C non-carrier	Atopy	2.727	0.586–12.690	NS
				0.573	0.192–1.706	NS
			Control	2.159	0.559–8.340	NS
				0.885	0.273–2.872	NS
	rs1544410 (BsmI) C > T	TT vs. CC CT vs. CC	Atopy	0.433	0.180–1.040	NS
				0.273	0.072–1.039	NS
			Control	1.907	0.923–3.944	NS
				2.209	0.866–5.635	NS
		C carrier vs. C non-carrier T carrier vs. T non-carrier	Atopy	0.438	0.106–9.474	NS
				0.544	0.269–1.232	NS
			Control	0.382	0.095–1.540	NS
				1.062	0.362–3.111	NS
	rs2228570 (FokI) G > A	AA vs. GG AG vs. GG	Atopy	1.732	0.768–3.907	NS
				3.800	1.190–12.134	0.024 *
			Control	2.092	1.044–4.194	0.037 *
				7.137	2.025–25.145	0.002 *
		G carrier vs. G non-carrier A carrier vs. A non-carrier	Atopy	1.552	0.326–7.386	NS
				0.280	0.098–0.804	0.018 *
			Control	2.159	0.559–8.340	NS
				0.229	0.069–0.761	0.016 *
	rs3847987 C > A	AA vs. CC AC vs. CC	Atopy	1.567	0.201–2.604	NS
				0.519	0.183–1.473	NS
			Control	35.997	6.401–202.446	<0.001 *
				4.727	1.489–15.007	0.008 *
		C carrier vs. C non-carrier A carrier vs. A non-carrier	Atopy	1.162	0.670–3.813	NS
				1.763	0.623–4.986	NS
			Control	1.253	0.286–2.264	NS
				4.522	1.429–14.308	0.010 *
	rs11168293 G > T	TT vs. GG GT vs. GG	Atopy	0.754	0.371–1.531	NS
				0.582	0.176–1.952	NS
			Control	1.138	0.578–2.243	NS
				1.969	0.696–5.525	NS
		G carrier vs. G non-carrier T carrier vs. T non-carrier	Atopy	0.765	0.225–2.595	NS
				1.731	0.569–5.269	NS
			Control	3.596	0.432–29.933	NS
				1.179	0.404–3.439	NS

* Significant difference. NS: not significant. OR: odds ratio. CI: Confidence interval.

**Table 6 biomolecules-14-00212-t006:** Distribution of VDR genotypes in atopic patients according to the total IgE and blood eosinophil levels.

SNP	Markers of Atopy	Genotypes *n* (%)			*p*-Value
rs731236 (TaqI) A > G	IgE level (IU/mL)	GG	AG	AA	
	>100	6 (9.1)	41 (62.1)	19 (28.8)	
	≤100	6 (11.5)	30 (57.7)	16 (30.8)	NS
	Eosinophils (%)				
	>5	4 (8.3)	35 (72.9)	9 (18.8)	
	≤5	8 (11.4)	36 (51.4)	26 (37.1)	0.054 *
rs7975232 (ApaI) A > C	IgE level (IU/mL)	AA	AC	CC	
	>100	16 (23.9)	35 (52.2)	16 (23.9)	
	≤100	15 (27.3)	26 (47.3)	14 (25.5)	NS
	Eosinophils (%)				
	>5	11 (22.4)	30 (61.2)	8 (16.3)	
	≤5	20 (27.4)	31 (42.5)	22 (30.1)	NS
rs1544410 (BsmI) C > T	IgE level (IU/mL)	CC	CT	TT	
	>100	26 (38.8)	34 (50.7)	7 (10.4)	
	≤100	25 (45.5)	25 (45.5)	5 (9.1)	NS
	Eosinophils (%)				
	>5	16 (32.7)	29 (59.2)	4 (8.2)	
	≤5	35 (47.9)	30 (41.1)	8 (11.0)	NS
rs2228570 (FokI) G > A	IgE level (IU/mL)	GG	AG	AA	
	>100	19 (28.4)	37 (55.2)	11 (16.4)	
	≤100	21 (38.2)	25 (45.5)	9 (16.4)	NS
	Eosinophils (%)				
	>5	16 (32.7)	26 (53.1)	7 (14.3)	
	≤5	24 (32.9)	36 (49.3)	13 (17.8)	NS
rs3847987 C > A	IgE level (IU/mL)	CC	AC	AA	
	>100	38 (56.7)	28 (41.8)	1 (1.5)	
	≤100	27 (49.1)	25 (45.5)	3 (5.5)	NS
	Eosinophils (%)				
	>5	27 (55.1)	20 (40.8)	2 (4.1)	
	≤5	38 (52.1)	33 (45.2)	2 (2.7)	NS
rs11168293 G > T	IgE level (IU/mL)	GG	GT	TT	
	>100	33 (49.3)	22 (32.8)	12 (17.9)	
	≤100	16 (29.1)	27 (49.1)	12 (21.8)	NS
	Eosinophils (%)				
	>5	25 (51.0)	19 (38.8)	5 (10.2)	
	≤5	24 (32.9)	30 (41.1)	19 (26.0)	0.046 *

* Significant difference. NS: not significant. OR: odds ratio. CI: confidence interval.

**Table 7 biomolecules-14-00212-t007:** Logistic regression analysis of the relationship between *VDR* SNPs in genotype and allele models and high total IgE levels (>100 IU/mL) in atopic patients.

Gene	SNP	Genotypic and Allelic Model	OR	95% CI	*p*-Value
*VDR*	rs731236 (TaqI) A > G	GG vs. AA	1.187	0.320–4.412	NS
		AG vs. AA	0.869	0.385–1.963	NS
		G carrier vs. G non-carrier	1.062	0.189–5.956	NS
		A carrier vs. A non-carrier	0.767	0.232–2.533	NS
	rs7975232 (ApaI) A > C	CC vs. AA	1.035	0.626–1.711	NS
		AC vs. AA	1.262	0.530–3.007	NS
		A carrier vs. A non-carrier	0.919	0.402–2.100	NS
		C carrier vs. C non-carrier	0.837	0.370–1.894	NS
	rs1544410 (BsmI) C > T	CC vs. TT	1.160	0.614–2.192	NS
		CT vs. TT	1.308	0.615–2.779	NS
		C carrier vs. C non-carrier	1.167	0.349–3.902	NS
		T carrier vs. T non-carrier	0.761	0.369–1.569	NS
	rs2228570 (FokI) G > A	AA vs. GG	1.162	0.670–1.992	NS
		AG vs. GG	1.636	0.734–2.646	NS
		G carrier vs. G non-carrier	1.004	0.383–2.631	NS
		A carrier vs. A non-carrier	0.641	0.300–1.371	NS
	rs3847987 C > A	AA vs. CC	0.487	0.153–1.550	NS
		AC vs. CC	0.796	0.383–1.653	NS
		C carrier vs. C non-carrier	0.263	0.027–2.599	NS
		A carrier vs. A non-carrier	1.359	0.664–2.781	NS
	rs11168293 G > T	TT vs. GG	0.696	0.423–1.147	NS
		GT vs. GG	0.395	0.174–0.898	0.027 *
		G carrier vs. G non-carrier	0.782	0.320–1.912	NS
		T carrier vs. T non-carrier	2.366	1.133–5.027	0.025 *

* Significant difference. NS: not significant. OR: odds ratio. CI: confidence interval.

**Table 8 biomolecules-14-00212-t008:** Logistic regression analysis of the relationship between *VDR* SNPs and peripheral blood eosinophils (>5%) in atopic patients.

Gene	SNP	Genotypic and Allelic Model	OR	95% CI	*p*-Value
*VDR*	rs731236 (TaqI) A > G	GG vs. AA	0.692	0.167–2.863	NS
		AG vs. AA	0.356	0.146–0.866	0.023 *
		G carrier vs. G non-carrier	0.319	0.163–0.934	0.035 *
		A carrier vs. A non-carrier	0.705	0.200–2.486	NS
	rs7975232 (ApaI) A > C	CC vs. AA	0.813	0.471–1.405	NS
		AC vs. AA	1.760	0.722–4.287	NS
		A carrier vs. A non-carrier	0.452	0.183–1.121	NS
		C carrier vs. C non-carrier	0.767	0.329–1.787	NS
	rs1544410 (BsmI) C > T	TT vs. CC	1.046	0.536–2.042	NS
		CT vs. CC	2.115	0.968–4.619	NS
		C carrier vs. C non-carrier	0.722	0.205–2.544	NS
		T carrier vs. T non-carrier	0.526	0.248–1.118	NS
	rs2228570 (FokI) G > A	AA vs. GG	0.899	0.515–1.570	NS
		AG vs. GG	1.083	0.482–2.433	NS
		G carrier vs. G non-carrier	0.769	0.283–2.091	NS
		A carrier vs. A non-carrier	0.990	0.458–2.141	NS
	rs3847987 C > A	AA vs. CC	0.740	0.428–3.259	NS
		AC vs. CC	0.853	0.406–1.793	NS
		C carrier vs. C non-carrier	1.511	0.206–11.099	NS
		A carrier vs. A non-carrier	1.130	0.547–2.337	NS
	rs11168293 G > T	TT vs. GG	0.503	0.285–0.886	0.017 *
		GT vs. GG	0.608	0.273–1.356	NS
		G carrier vs. G non-carrier	0.323	0.112–0.935	
		T carrier vs. T non-carrier	2.127	1.012–4.471	

* Significant difference. NS: not significant. OR: odds ratio. CI: confidence interval.

**Table 9 biomolecules-14-00212-t009:** The odds ratio for increased IgE and blood eosinophils in atopy.

	Parameters	OR	95% CI	*p*-Value
Total IgE > 100 IU/mL	Vitamin D level < 30 ng/mL	3.123	0.963–10.124	0.058
Eosinophil > 5%		1.167	0.373–3.650	NS
Total IgE > 100 IU/mL	Vitamin D level < 30 ng/mL + rs11168293 GG genotype	2.541	1.142–5.656	0.022 *
Eosinophil > 5%		2.267	1.042–4.933	0.039 *
Total IgE > 100 IU/mL	Vitamin D level < 30 ng/mL + rs11168293 GG genotype + rs731236 G allele	0.701	0.299–1.646	NS
Eosinophil > 5%		0.316	0.123–0.815	0.017 *

* Significant difference. NS: not significant. OR: odds ratio. CI: confidence interval.

## Data Availability

All data generated or analyzed during this study are included in this published article.

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
