# Peer review of "The Association of Vitamin D Receptor Gene Polymorphisms with Vitamin D, Total IgE, and Blood Eosinophils in Patients with Atopy"

_biomolecules, 2024, doi:10.3390/biom14020212_

Round 1
Reviewer 1 Report
Comments and Suggestions for Authors
The manuscript can be published now.
Author Response
We express our sincere gratitude to the reviewer for dedicating their valuable time to a thorough review of the manuscript.
Reviewer 2 Report
Comments and Suggestions for Authors
This is a cross-sectional case -control study on the associations of VDR polymorphisms with 25(OH)D concentrations in atopic disease compared with controls.
The study lacks a clear research hypothesis:
1) Do specific VDR polymorphisms are associated with atopic outcomes through their effects on 25(OH)D (that means the authors hypothesize that specifix SNPs affect 25-hydroxyvitamin D and results in higher rates of hypovitaminosis D and consequently to aggravation of atopy) -that means you evaluate 25(OH)D in both cases and controls (not just cases as in this study) and elucidate on associations of VDRs SNPs in both groups
OR
2) VDR polymorphisms are indepedently associated with atopic outcomes (irrespective of 25-OHD) -That means you study vitamin D sufficient populations (both cases and coltrols) and focus on potential associations
Neither of the above has been evaluated in addition to lack of adjustment for seasonal 25(OH)D variation (i.e. would the results be significant in another season ),vitamin D nutritional intake,BMI etc.
Lack of the same analysis for controls as conducted for cases for 25(OH)D and VDRs could reveal similar significant results and largely not apply for atopic patients.I would strongly suggest evaluating VDR and 25 (OH)D tertile associations in controls as well, otherwise the study design is poor.
CVs of assays used as well as other atopic outcomes (inhaler use, hospitalization ) not reported
Comments on the Quality of English LanguageExtensive language editing is required
